# Lung evolution in vertebrates and the water-to-land transition

Camila Cupello[1]*, Tatsuya Hirasawa[2], Norifumi Tatsumi[3], Yoshitaka Yabumoto[4], Pierre Gueriau[5,6], Sumio Isogai[7], Ryoko Matsumoto[8], Toshiro Saruwatari[9,10], Andrew King[11], Masato Hoshino[12], Kentaro Uesugi[12], Masataka Okabe[3], Paulo M Brito[1]*

[1]Departamento de Zoologia-IBRAG, Universidade do Estado do Rio de Janeiro, Rio de Janeiro RJ, Brazil; [2]Department of Earth and Planetary Science, Graduate School of Science, The University of Tokyo, Tokyo, Japan; [3]Department of Anatomy, The Jikei University School of Medicine, Tokyo, Japan; [4]Kitakyushu Museum of Natural History and Human History, 2-4-1 Higashida, Yahatahigashi-ku, Kitakyushu, Fukuoka, Japan; [5]Institute of Earth Sciences, University of Lausanne, Lausanne, Switzerland; [6]Université Paris-Saclay, CNRS, ministère de la Culture, UVSQ, MNHN, Institut photonique d'analyse non-destructive européen des matériaux anciens, Saint-Aubin, France; [7]Department of Anatomy, Iwate Medical University School of Medicine, Iwate, Japan; [8]Kanagawa Prefectural Museum of Natural History, Kanagawa, Japan; [9]Atmosphere and Ocean Research Institute, The University of Tokyo, Chiba, Japan; [10]Seikei Education and Research Center for Sustainable Development, Tokyo, Japan; [11]Synchrotron SOLEIL, L'orme des Merisiers Saint-Aubin, Gif-sur-Yvette Cedex, France; [12]Japan Synchrotron Radiation Research Institute (JASRI/SPring-8), Hyogo, Japan

*For correspondence:
camila.dc@gmail.com (CC);
pbritopaleo@yahoo.com.br
(PMB)

Competing interest: The authors declare that no competing interests exist.

**Abstract** A crucial evolutionary change in vertebrate history was the Palaeozoic (Devonian 419–359 million years ago) water-to-land transition, allowed by key morphological and physiological modifications including the acquisition of lungs. Nonetheless, the origin and early evolution of vertebrate lungs remain highly controversial, particularly whether the ancestral state was paired or unpaired. Due to the rarity of fossil soft tissue preservation, lung evolution can only be traced based on the extant phylogenetic bracket. Here we investigate, for the first time, lung morphology in extensive developmental series of key living lunged osteichthyans using synchrotron x-ray microtomography and histology. Our results shed light on the primitive state of vertebrate lungs as unpaired, evolving to be truly paired in the lineage towards the tetrapods. The water-to-land transition confronted profound physiological challenges and paired lungs were decisive for increasing the surface area and the pulmonary compliance and volume, especially during the air-breathing on land.

## Editor's evaluation

This study focused on five osteichthyan vertebrate species and investigated their lung morphology. The comparison of the observations suggests an origin of the lung as an unpaired organ, with the present-day paired forms in amniotes being a result of secondary modification. The sound morphological comparison presented provides valuable insight into the evolution of the lung. The work will be of interest to colleagues studying vertebrate evolution.

**eLife digest** All life on Earth started out under water. However, around 400 million years ago some vertebrates, such as fish, started developing limbs and other characteristics that allowed them to explore life on land. One of the most pivotal features to evolve was the lungs, which gave vertebrates the ability to breathe above water.

Most land-living vertebrates, including humans, have two lungs which sit on either side of their chest. The lungs extract oxygen from the atmosphere and transfer it to the bloodstream in exchange for carbon dioxide which then gets exhaled out in to the atmosphere. How this important organ first evolved is a hotly debated topic. This is largely because lung tissue does not preserve well in fossils, making it difficult to trace how the lungs of vertebrates changed over the course of evolution.

To overcome this barrier, Cupello et al. compared the lungs of living species which are crucial to understand the early stages of the water-to-land transition. This included four species of lunged bony fish which breathe air at the water surface, and a four-legged salamander that lives on land.

Cupello et al. used a range of techniques to examine how the lungs of the bony fish and salamander changed shape during development. The results suggested that the lungs of vertebrates started out as a single organ, which became truly paired later in evolution once vertebrates started developing limbs. This anatomical shift increased the surface area available for exchanging oxygen and carbon dioxide so that vertebrates could breathe more easily on land.

These findings provide new insights in to how the lung evolved into the paired structure found in most vertebrates alive today. It likely that this transition allowed vertebrates to fully adapt to breathing above water, which may explain why this event only happened once over the course of evolution.

## Introduction

Lungs, the most important organ of the pulmonary complex, are rarely preserved in fossils, hindering direct evidence of how the earliest air-breathing vertebrates breathed air. Indeed, among fossil taxa, the presence of lungs has only been confirmed in coelacanths (*Brito et al., 2010*; *Cupello et al., 2017b*; *Cupello et al., 2019*), a salamander (*Tissier et al., 2017*), an ornithuromorph bird (*Wang et al., 2018*), and possibly a tadpole (*Rossi et al., 2019*). In addition, the presence of specialized skeletal structures, such as cranial ribs and enlarged spiracular openings, provided indirect evidence of air-breathing behaviour in a wider range of fossil sarcopterygians including lungfishes and early tetrapods, but also actinopterygians (*Long, 1993*; *Long et al., 2006*; *Clement and Long, 2010*; *Clement et al., 2016*; *Cupello et al., 2019*).

Altogether, the evolutionary origin of the vertebrate lung can therefore be narrowed down to osteichthyans (*Goujet, 2011*; *Tatsumi et al., 2016*). The interpretation of thoracic paired masses in the antiarch placoderm *Bothriolepis canadensis* as a lung (*Denison, 1941*; *Arsenault et al., 2004*) even suggested an origin among early jawed vertebrates, dating back to at least the Devonian period. Yet, lung affinities for such structures remain elusive (*Janvier et al., 2007*) and could not be confirmed by anatomical, phylogenetic, or biological data (*Goujet, 2011*; *Béchard et al., 2014*). Here, we follow *Janvier et al., 2007*, *Goujet, 2011* and *Béchard et al., 2014*, and consider that observable evidences are elusive and do not support the interpretation of these paired masses as a lung.

Our knowledge about the morphological and genetic development of the lung is, however, highly biased towards amniotes, and consequently the original form of this evolutionary novelty among osteichthyans remains largely elusive. One hypothesis, formed and supported by studies on tetrapods (particularly mammals and birds), assumes that the lung evolved through a modification of the pharyngeal pouch (*Kastschenko, 1887*), as the lung bud develops at the pharyngo-oesophageal junction during embryonic development. Consequently, this view (*Kastschenko, 1887*; *Kuratani and Tanaka, 1990*) predicts that the primitive lungs appeared as bilaterally paired organs at the caudolateral part of the pharynx. Indeed, in embryology, lungs of living tetrapods have been mostly described as paired derivates of the respiratory tube, arisen from paired and small hollow swellings (*Flint, 1906*). Previous studies on amphibians have also proposed that the lung bud develop from paired rudiments of the ventral portions of the eighth pharyngeal pouches (*Goodrich, 1931*; *Marcus, 1937*; *Perry et al., 2001*). Additionally, the plesiomorphic state of lungs has been mostly described as paired organs (*Funk et al., 2020*). On the other hand, another hypothesis does not constrain the evolutionary origin

of the lung to the serial homologue of the pharyngeal pouch (*Greil, 1913*; *Neumayer, 1930*; *Wassnetzov, 1932*). In this view, although the possibility that the primitive lung developed on the pharyngeal endoderm is not excluded, the primitive lung is considered to appear on the floor of the pharynx, or more generally, on the floor of the foregut. This scenario does not predict bilaterally paired forms of primitive lungs.

Curiously, some living vertebrates display an unpaired organ, leaving the ancestral condition equivocal. The sister group to all other extant actinopterygians, the obligate air-breathing polypterids (*Icardo et al., 2017*), breath air using lungs, which have previously been described as a paired organ (*Icardo et al., 2017*; *Geoffrey Saint Hilaire, 1802*; *Graham, 1997*). However, in adult specimens of *Polypterus senegalus* the glottis only opens to the right sac and the left sac is connected to the right sac by a separate opening (*Graham, 1997*), raising old questions about its true paired condition. Among sarcopterygians, the unpaired lung of coelacanths is unequivocal. The living coelacanth *Latimeria chalumnae*, that inhabits moderate deep-water and makes gas exchanges only through gills, have an unpaired lung with no outline of a second bud at different developmental stages (*Cupello et al., 2015*; *Cupello et al., 2017a*; *Cupello et al., 2017b*; *Cupello et al., 2019*). In the extant sister group of all tetrapods (*Amemiya et al., 2013*), namely lungfishes, the three extant genera have lungs capable to uptake oxygen from the air. However, in the facultative air-breather *Neoceratodus forsteri*, the lung is described as unpaired in adults (*Greil, 1913*; *Graham, 1997*; *Grigg, 1965*). In contrast with both South American and African lungfishes, the Lepidosirenoidea, that are obligated air-breathers and have a lung described as a ventral paired organ (*Graham, 1997*) like tetrapod lungs.

To follow lung evolutionary history in vertebrates, we analyzed the morphogenesis of lungs of key living osteichthyans (*Figures 1 and 2*, *Figures 3 and 4*, *Figures 5 and 6*). Embryos, larvae, juveniles and adults of *P. senegalus*, *N. forsteri*, *Lepidosiren paradoxa* were examined. To compare the lung anatomy of osteichthyan fishes with tetrapods, we studied also an extensive developmental series of the living salamandrid *Salamandra salamandra*, from early and late larvae to juveniles before and after metamorphosis (*Figure 5*). As salamanders are often considered to have retained plesiomorphic characteristics of tetrapod stance and locomotion (*Pierce et al., 2020*), we used them here also as a model for understanding lung evolution in tetrapods. Specimens of mentioned taxa were examined through x-ray microtomography, the unique effective non-invasive methodology to study their morphology and histology at a three-dimensional (sub) microscale. When possible, we proceeded also with dissections and the study of histological sections. We compare our results with the available information from the lung of fossil taxa, the coelacanths and salamanders (*Cupello et al., 2019*; *Brito et al., 2010*; *Tissier et al., 2017*).

## Results
### The lung development in *Polypterus senegalus*

From our observations on embryos of 8.0, 8.5, 9.1, and 9.3 mm total length (TL), the lung anlage develops as a ventral unpaired and tubular depression of the respiratory pharynx (the posterior portion of the pharynx), surrounded by undifferentiated mesenchymal cells (*Tatsumi et al., 2016*; *Figures 1A, B, 2B and C*). Only at the 12 mm TL larva, the left bud arises from the principal and primary lung anlage as a branch (*Figure 2D*). Subsequently, the lung assumes its asymmetrical morphology, the left sac is smaller and remains ventral in the abdominal cavity, while the right one (principal tube) starts a partial turn up and stays parallel to the dorsal portion of the foregut (including the stomach). The left sac keeps a secondary connection to a lateral opening of the principal tube, and not to the foregut (*Figures 1D, E, 6A and B*, *Figure 1—figure supplements 1 and 2*). Undifferentiated dense cells surrounding the glottis are visible for the first time in specimens of 15.5 mm TL (black arrows in *Figure 2E*). Air-breathing behavior starts at the juvenile stage in *P. senegalus* (*Bartsch et al., 1997*), and from juveniles of 23 mm TL onward, the blastema starts to develop into the muscular sphincter and respiratory epithelium at the glottis (ciliated cells intercalated by goblet cells). Right and left sacs are well developed and have a projection anterior to the connection with the small pneumatic duct in juveniles (*Figure 1D and E*). The right tube is three times longer than the left one, with an expanded diameter in its caudal portion, posterior to the stomach (*Figure 1C and D*).

The right and left sacs make a partial turn-up, remaining parallel to the dorsal surface of the upper gastrointestinal tract (one of each side). *P. senegalus* lung is internally smooth and lacks alveolation at

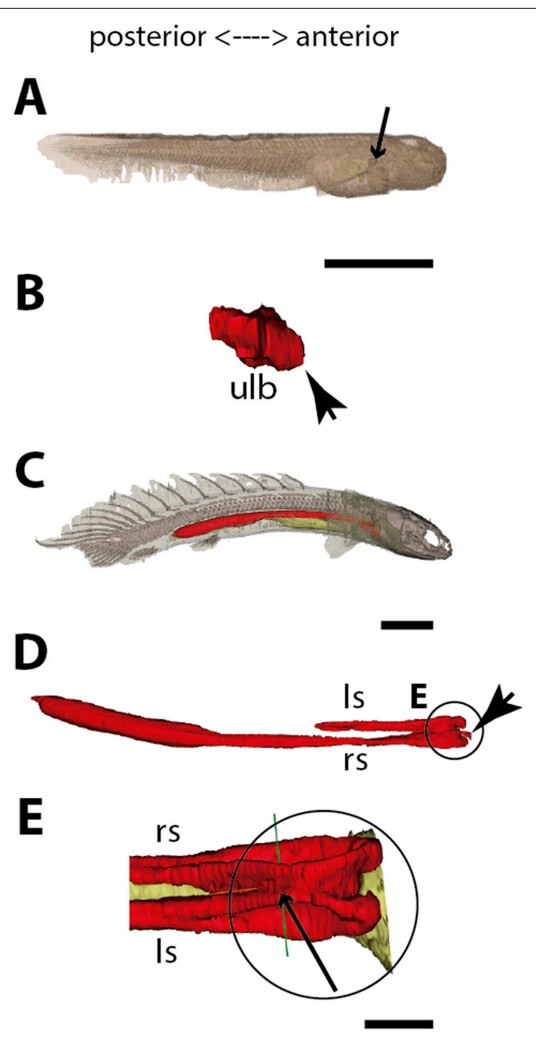

**Figure 1.** Three-dimensional reconstructions of the pulmonary complex of *Polypterus senegalus*. (**A**) Early embryo (9.3mm) total length (TL) in right lateral view, (**B**) isolated lung of the early embryo in dorsal view, (**C**) juvenile (45mm TL) in right lateral view, (**D**) isolated lung of the juvenile in dorsal view, (**E**) close-up of (**D**) highlighting the lung in ventral view and pointing out the region of the independent and secondary connection of the left sac to the right one by a lateral opening. Yellow, foregut including the stomach; red, lung. Black arrow in (**A**) pointing to the lung. Arrowheads in (**B**) pointing to the lung connection to the foregut and in (**D**) pointing to the pneumatic duct connection to the foregut. Black arrow in (**E**) pointing to the independent connection. Ls, left sac; rs, right sac; ulb, unpaired lung bud. Scale bars, 5.0mm (**A**); 0.075mm (**B**); 5.0mm (**C, D**); 1.0mm (**E**).

The online version of this article includes the following figure supplement(s) for figure 1:

**Figure supplement 1.** Sections of synchrotron x-ray microtomography of a juvenile of *Polypterus senegalus* (23 mm total length; TL).

**Figure supplement 2.** Three-dimensional reconstructions of the pulmonary complex of *Polypterus senegalus*.

all the examined developmental stages, except for in the 45 mm TL juvenile, in which the most anterior projection of the lung, anterior to the connection with the pneumatic duct, is slightly compartmentalized. This evidence based on the first developmental stages of *P. senegalus* (embryos with 8.5 mm, 9.1 mm, and 9.3 mm) lung prove that the lung bud initially develops as an unpaired anlage in this taxon (*Figures 1B, 2B and C*). The left sac develops secondarily from the right sac at later developmental stages, as a diverticulum or a lobe, of the right primary lung (*Figure 2E and F*, *Figure 1— figure supplements 1 and 2*).

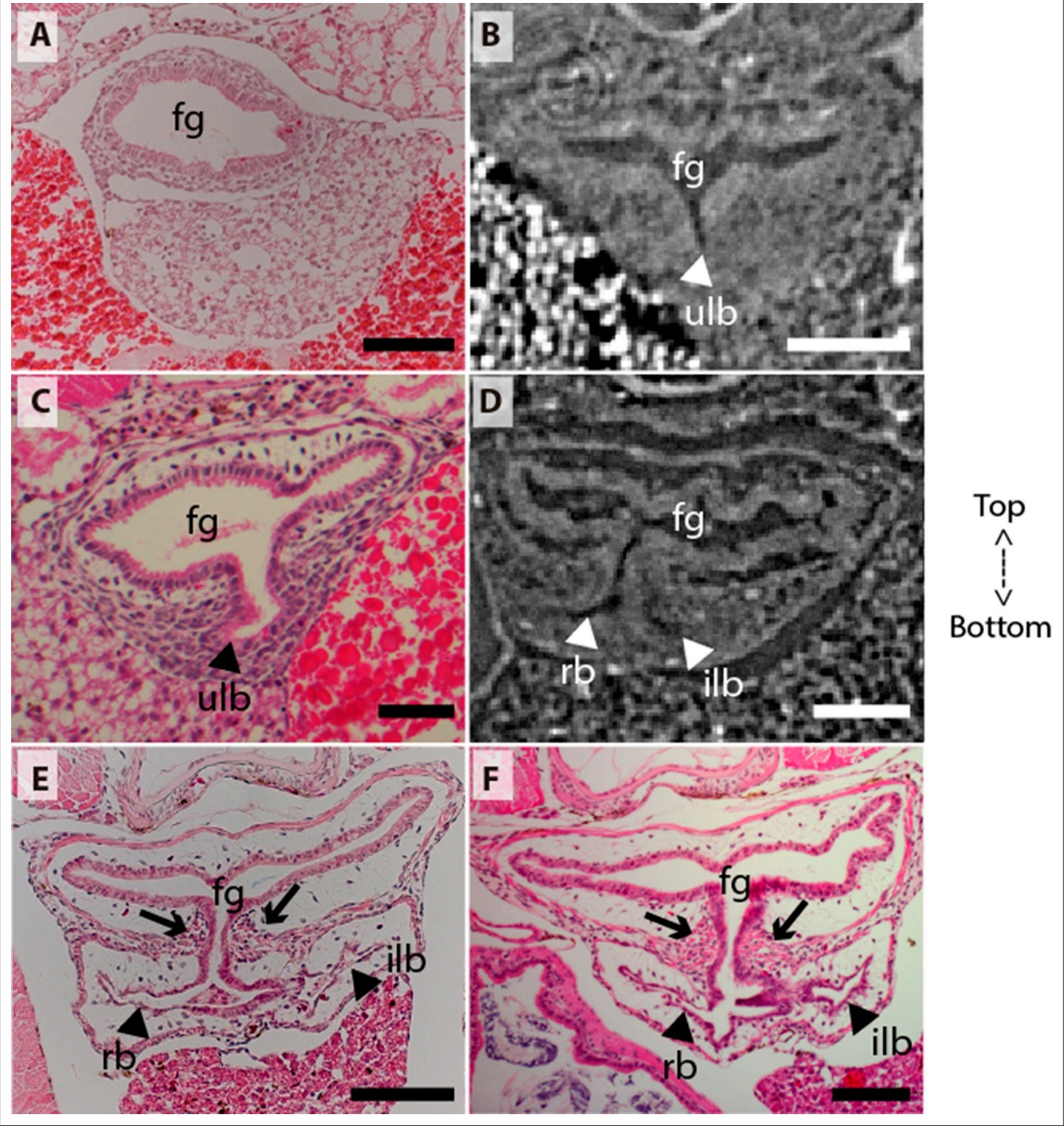

**Figure 2.** Coronal sections of the unpaired lung in the living actinopterygian fish *Polypterus senegalus*. (**A**) No lung bud in 8.0mm total length (TL) specimen, (**B**) origin of an unpaired lung bud in 8.5mm TL specimen, (**C**) unpaired lung bud in 9.1mm TL specimen, (**D**) first register of an independent and lateral second lung bud in 12mm TL specimen, (**E, F**) independent and lateral second lung bud arising from the principal tube in 15.5mm TL and 18mm TL specimens. (**A, C, E–F**) Histological thin-sections. (**B, D**) Sections of synchrotron x-ray microtomography of the early embryo. Black and white head arrows pointing to the lumen of the unpaired lung buds; arrows pointing to the undifferentiated cells surrounding the glottis. Fg, foregut; ilb, independent lateral bud; rb, right bud; ulb, unpaired lung bud. Scale bars, 0.2mm (**A, E**); 0.1mm (**B, D, F**); 0.05mm (**C**).

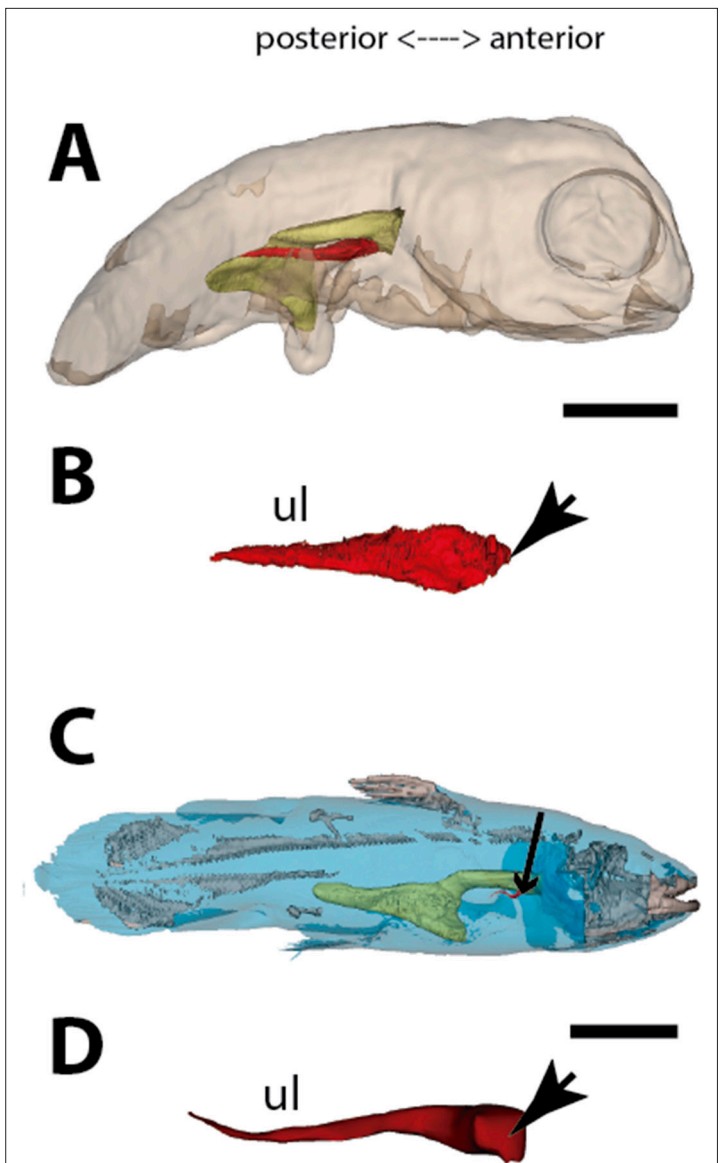

**Figure 3.** Three-dimensional reconstructions of the pulmonary complex of *Latimeria chalumnae*. (**A**) Early embryo of *Latimeria chalumane* (45mm total length; TL) in right lateral view (*Cupello et al., 2015*), (**B**) isolated unpaired lung of the early embryo in dorsal view, (**C**) adult specimen of *Latimeria chalumnae* (1300mm TL) in right lateral view (*Cupello et al., 2015*), (**D**) isolated unpaired lung of the adult specimen in dorsal view. Yellow, foregut including the stomach; red, lung. Arrowheads in (**B**) and (**D**) pointing to the lung connection to the foregut. Black arrow in (**C**) pointing to the lung. Ul, unpaired lung bud in (**B**) and unpaired lung in (**D**). Scale bars, 5.0mm (**A**); 5.0mm (**B**); 200.0mm (**C**); 40mm (**D**). Modified from *Cupello et al., 2015*.

## The lung development in *Latimeria chalumnae*

The anatomy of the lung of fossil and extant coelacanths, as well as its ontogenic development in the extant *L. chalumnae*, have been extensively documented (*Cupello et al., 2015*; *Cupello et al., 2017a*; *Cupello et al., 2017b*; *Cupello et al., 2019*). The latter studies highlighted that embryos of *L. chalumnae* display ventral compartmentalized unpaired lung throughout its length, suggesting alveolation (*Cupello et al., 2017a*), and that a lateral and internal chamber is also present in the early embryo (40 mm TL) (*Figures 3 and 6D*). At the latest developmental stages the pulmonary complex shows vestigial features, and no internal compartmentalization is recognizable (*Lambertz et al., 2015*). Adult specimens have constrictions and septations that divide the unpaired lung into separate lobes throughout its length, as in some fossil coelacanths (*Cupello et al., 2019*). Fossil coelacanths,

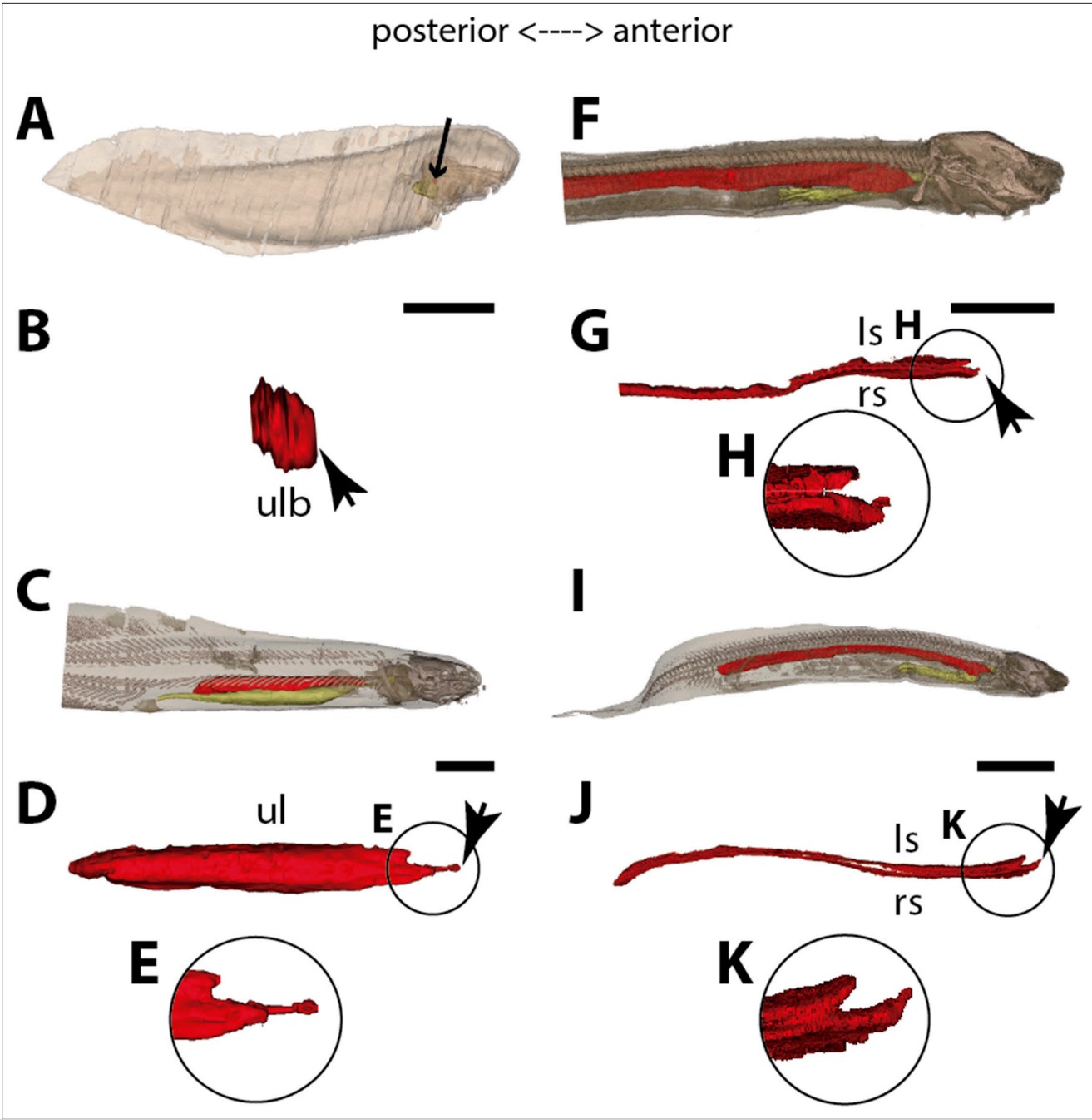

**Figure 4.** Three-dimensional reconstructions of the pulmonary complex of two species of lungfishes. (**A**) Early embryo of *Neoceratodus forsteri* (13.5mm total length; TL) in right lateral view, (**B**) isolated unpaired lung of the early embryo in dorsal view, (**C**) adult specimen of *Neoceratodus forsteri* (200mm TL) in right lateral view, (**D**) isolated unpaired lung of the adult specimen in dorsal view, (**E**) close-up of the lung unpaired connection to the foregut in (**D**), (**F**) larva of *Lepidosiren paradoxa* (46mm TL) in lateral view, (**G**) isolated lung of the larval specimen in dorsal view, (**H**) close-up of the lung unpaired connection to the foregut in (**G**), (**I**) juvenile of *Lepidosiren paradoxa* young adult (68mm TL) in lateral view, (**J**) isolated lung of the juvenile specimen in dorsal view, (**K**) close-up of the lung unpaired connection to the foregut in (**J**).Yellow, foregut including the stomach; red, lung. Black arrow in (**A**) pointing to the lung. Arrowheads in (**B**), pointing to the lung connection to the foregut and in (**D**), (**G**) and (**J**) pointing the pneumatic duct connection to the foregut. Ls, left sac; rs, right sac; ul, unpaired lung; ulb, unpaired lung bud. Scale bars, 2.5mm (**A**); 0.1mm (**B**); 20mm (**C**); 10mm (**D, I**); 7.2 mm (**J**); 5.0mm (**F, G**).

The online version of this article includes the following figure supplement(s) for figure 4:

**Figure supplement 1.** Sections of synchrotron x-ray microtomography of a larva of *Neoceratodus forsteri* (19 mm total length; TL).

**Figure supplement 2.** Sections of synchrotron x-ray microtomography of a juvenile of *Lepidosiren paradoxa* (68mm total length; TL).

**Figure supplement 3.** Dissection of the lung of an adult *Lepidosiren paradoxa* (400mm total length; TL).

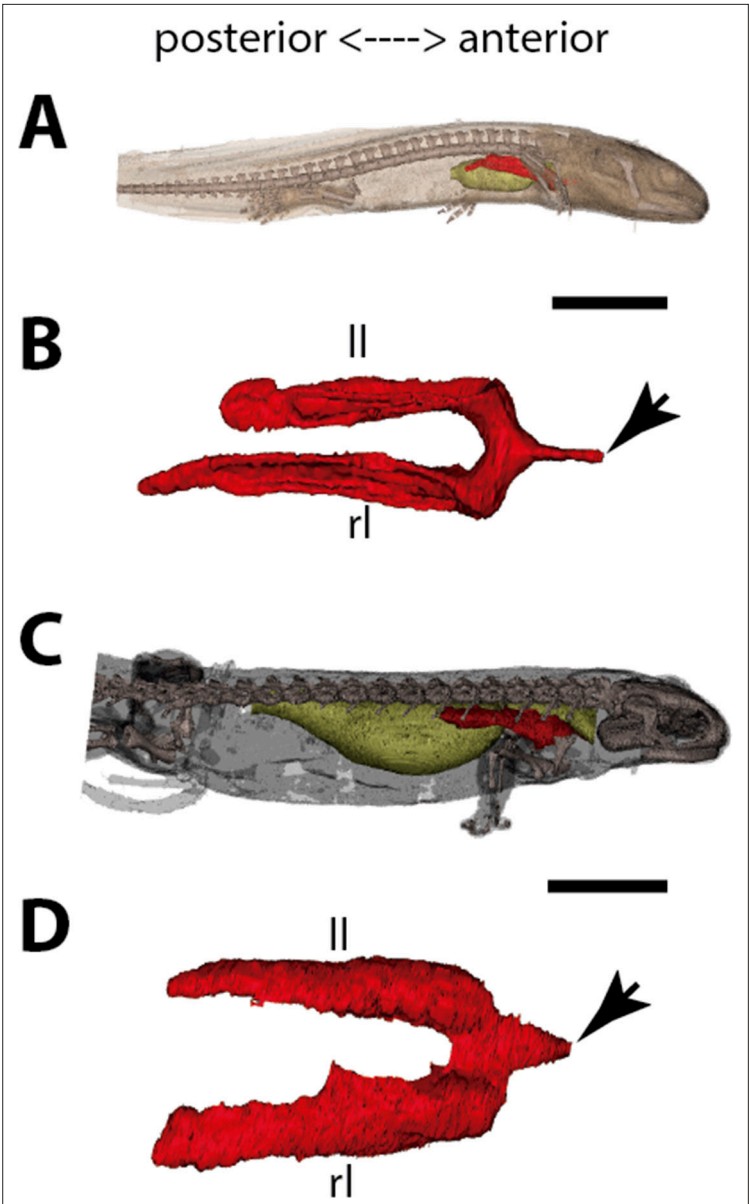

**Figure 5.** Three-dimensional reconstructions of the pulmonary complex of *Salamandra salamandra*. (**A**) Early larva of *Salamandra salamandra* (35.5mm total length; TL) in right lateral view, (**B**) isolated paired lung of the larva embryo in dorsal view, (**C**) juvenile of *Salamandra salamandra* (81.85mm TL) in right lateral view, (**D**) isolated paired lung of the juvenile specimen in dorsal view. Yellow, foregut including the stomach; red, lung. Arrowheads in (**B**) and (**D**) pointing to the trachea connection to the foregut. Ll, left lung; rl, right lung. Scale bars, 5.0mm (**A**); 3.125mm (**B**); 10mm (**C**); 6.25cm (**D**).

The online version of this article includes the following figure supplement(s) for figure 5:

**Figure supplement 1.** Sections of synchrotron x-ray microtomography of a larva of *Salamandra salamandra* (42.8mm total length; TL).

from late Devonian to late Cretaceous, were most probably facultative air-breathers and made gas exchanges through their unpaired lungs and gills (*Cupello et al., 2019*; *Brito et al., 2010*). Although some authors suggest that *L. chalumnae* fatty organ evidences a paired lung, previous studies proved that this organ is not the second lung, since there is no opened connection between this organ and the foregut or lung, nor lung plates surrounding it (*Cupello et al., 2015*; *Cupello et al., 2017a*; *Cupello et al., 2017b*). Based on these, the paired condition of coelacanth lungs can be excluded.

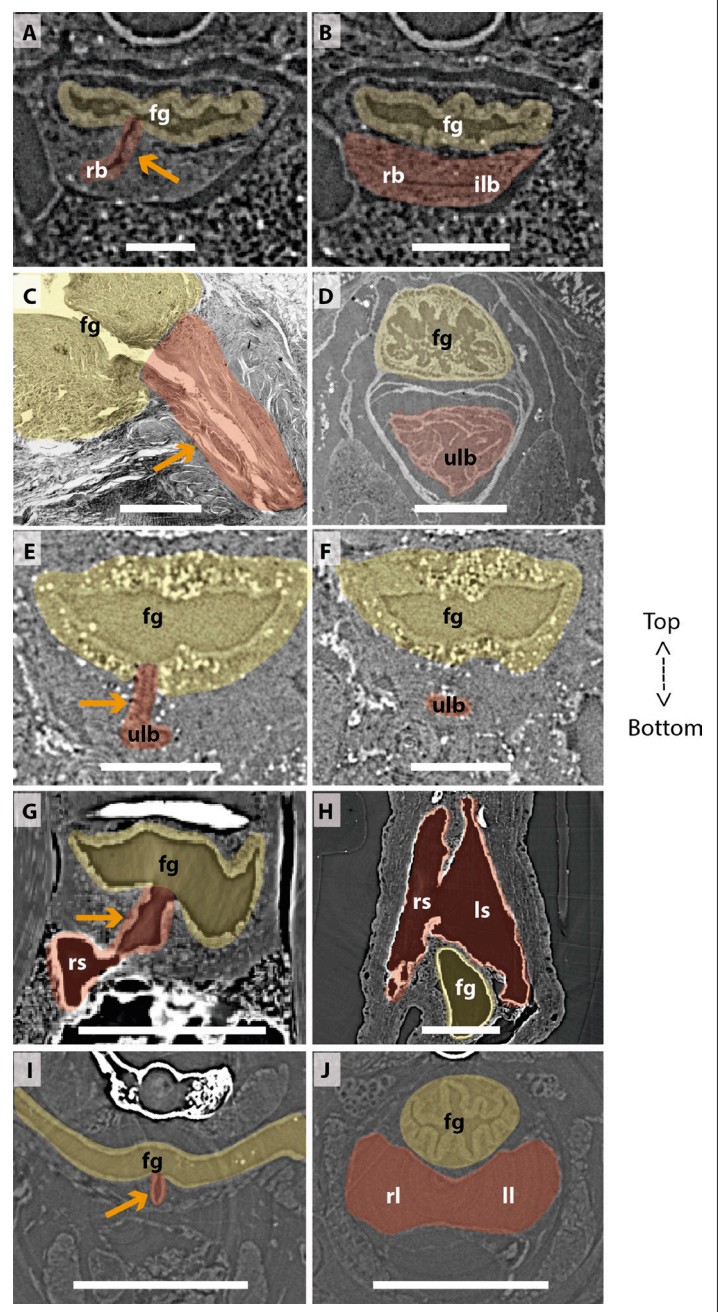

**Figure 6.** Comparison of sections showing the difference in lung origin and connection between unpaired (**A–H**) and true paired lungs (**I, J**). (**A, B**) Virtual section of *Polypterus senegalus* (12mm total length; TL), (**C**) histological thin section of *Latimeria chalumnae* (127cm) (***Cupello et al., 2017a***), (**D**) virtual section of *L. chalumnae* (40mm TL; modified from ***Cupello et al., 2017a***), (**E, F**) virtual section of *Neoceratodus forsteri* (16mm TL), (**G, H**) virtual section of *Lepidosiren paradoxa* (46mm TL), (**I, J**) virtual section of *Salamandra salamandra* (35.5mm TL). Yellow, foregut including the stomach; red, lung. Orange arrows, opened connection between the foregut and the lung. Fg, foregut; ll, left lung; ls, left sac; rb, right bud; ilb, independent lung bud; rl, right lung; rs, right sac; ulb, unpaired lung bud. Scale bars, 0.25mm (**A, B**); 3.0mm (**C**); 1.0mm (**D**); 0.1mm (**E, F**); 0.5mm (**G, H**); 1.25mm (**I, J**).

## The lung development in *Neoceratodus forsteri*

The first developmental stage with lung anlage registered in this study is an early larva of 13.5 mm TL, with an unpaired morphology represented primarily by lung anterior projection (***Figure 4B***). In larvae of 16 mm, 17 mm, and 17.5 mm TL, although organogenesis is still not complete, a long and

unpaired lung anlage is clearly identifiable and arises as a ventral depression of the post-pharyngeal foregut (*Figure 6E and F*). In the 19 mm TL specimen, the unpaired lung starts its dorsal turn up in relation to the dorsal portion of the foregut (including the stomach) (*Figure 4—figure supplement 1*). From 20.5 mm TL onward, organogenesis is completed. In the larva of 20.5 mm TL, the lung remains unpaired and is ventrally connected to the post-pharyngeal foregut by a ventral, opened and long pneumatic duct. This organ has a projection anterior to the connection of the pneumatic duct and does not display alveolation/compartmentalization yet. According to previous studies (*Kemp, 1982*; *Kemp, 1986*), air-breathing begins in *N. forsteri* at 25 mm TL larval stage. Our results reveal that the larva with 26.5 mm TL presents a lung wall slightly pleated. From 50 mm TL larval stage onward, the lung wall is pleated eventually providing a high degree of alveolation. In the adult individual with 200 mm TL, the single lung displays internally two lateral chambers that are connected to a principal median chamber at both sides as a single structure (*Figure 4C–E*). In adult specimens of *N. forsteri*, the lung is highly compartmentalized by septa of smooth muscle and non-elastic connective tissue, as well as spongy alveolar structures (*Grigg, 1965*). At this developmental stage, the lung makes a complete dorsal turn-up at its posterior portion in relation to the gastrointestinal tract (*Figure 4C*). Although some authors pointed the presence of a second bud at the early developmental stages of *N. forsteri*, referred as the left lung (*Spencer, 1893*; *Neumayer, 1904*), the results presented here show an indubitably unpaired configuration for *Neoceratodus* lung throughout the ontogeny (*Figures 4A–E, 6E and F* and *Figure 4—figure supplement 1*).

## The lung development in *Lepidosiren paradoxa*

Lungs of the four specimens studied here, from larva to adults (larva with 46 mm TL, juveniles with 68 mm TL and 222.1 mm TL, and adult with 400 mm TL), display a similar morphology, and surprisingly, left and right tubes do not arise simultaneously. Only the right sac is connected to the pharynx by a long pneumatic duct (*Figure 6G and H* and *Figure 4—figure supplement 2*). The left sac is a branch of the right one, connected by a posterior and secondary opening at the lung level, already in dorsal position in relation to the foregut (*Figure 6G and H* and *Figure 4—figure supplement 2*). There is no connection of the left sac with the pneumatic duct. In *L. paradoxa*, only the pneumatic duct is ventrally positioned, and the lung makes a complete dorsal turn up from the right side of the upper gastrointestinal tract (*Figure 4F–K* and *Figure 4—figure supplement 3*), just after the ventral connection to the pharynx. This complete dorsal turn-up is also seen in the lung of adult specimens of *N. forsteri* (*Figure 4I–K*). There are no anterior projections of the lung. Lung compartmentalization is clearly observable through dissections, evidencing the high degree of alveolation (*Figure 4—figure supplement 3*). Our results indicate that the lung of *L. paradoxa* is, in fact, remarkably similar to *P. senegalus* lung. The so-called left lung of *L. paradoxa* is most likely a diverticulum or a modified lateral lobe, which might have evolved secondarily, an advantage for enlarging the surface area for oxygen-uptake, eventually enabling the obligatory air-breathing performance in the linage towards *L. paradoxa*.

## The lung development in *Salamandra salamandra*

In early larvae with 35.5 mm TL and 42.8 mm TL, paired lungs are collapsed in its middle and posterior portion (*Figure 5A and B*). The internal lung wall is thin and smooth, without compartmentalization and/or alveolation in its inner wall (*Figure 6J* and *Figure 5—figure supplement 1*). From the early larvae onward, the muscular glottis develops on the ventral portion of the pharynx, and both left and right lungs arise simultaneously and symmetrically from a long trachea and paired first order bronchioles (*Figures 5C, D, 6I and J* and *Figure 5—figure supplement 1*). Lungs are symmetrical in size and morphology and are placed in the anteriormost portion of the abdominal cavity, as described for other tetrapods (*Figure 5C and D*).

In postmetamorphic juveniles (of 81.85 mm TL), paired lungs are already functional, not collapsed, and the main organ for oxygen-uptake (*Goniakowska-Witalińska, 1978*; *Goniakowska-Witalińska, 1982*). From this developmental stage onward, lungs are highly compartmentalized by multiple septa. Due to the paired and compartmentalized anatomy, the lung surface area for oxygen-uptake, as well as its volume capacity, increase substantially – both important features for a functional lung in dry environments. Here we confirm that at different developmental stages of *S. salamandra*, lungs are truly paired since both left and right lungs arise simultaneously and symmetrically and are directly

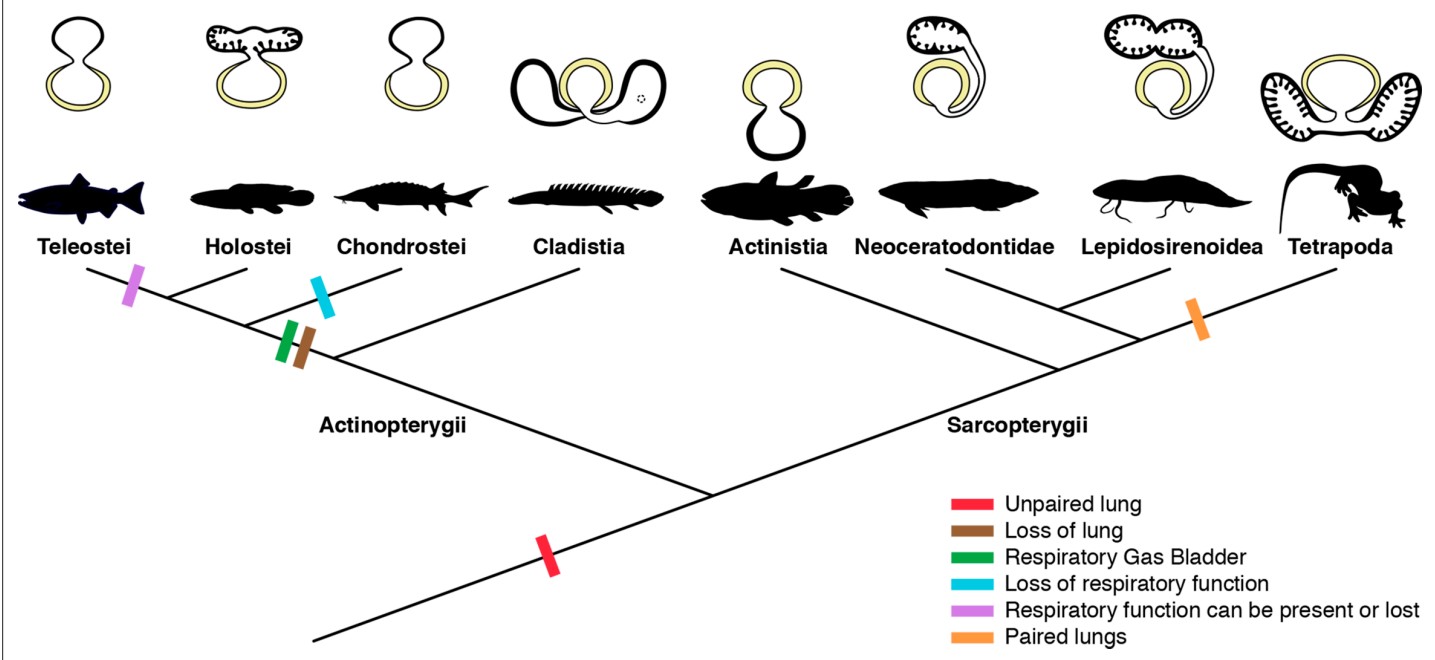

**Figure 7.** Schematic figure reconstructing the evolutionary history of vertebrate lungs. All living actinopterygian and sarcopterygian fishes have unpaired lungs. True paired lungs are a synapomorphy of tetrapods. Dashed circle in Cladistia lung pointing to the secondary and independent opening to a left sac, at the lung level. Modified from *Liem, 1988*. This figure was made with free silhouettes from PhyloPic.

connected to the trachea. Throughout the ontogeny, *S. salamandra* lungs have a ventral origin, and makes a partial dorsal turn-up in its posterior portion, remaining parallel to the dorsal wall of the upper gastrointestinal tract. Due to the rarity of soft tissue preservation in the fossil record, only one species of salamandrid, *Phosphotriton sigei* presents its lung preserved (*Tissier et al., 2017*). This lung is described as multichambered, placed in the anteriormost part of the abdominal cavity (*Tissier et al., 2017*), such as in the living salamandrid described above.

## Discussion

Traditionally, vertebrate lungs are defined as ventral paired organs derived from the ventral portion of the posterior pharynx or post-pharyngeal foregut (*Perry et al., 2001*; *Funk et al., 2020*; *Lambertz et al., 2015*; *Graham, 1997*; *Kardong, 2015*). However, we demonstrate here the presence of an unambiguous unpaired lung, that develops from the ventral foregut, but sometimes occupy the dorsal position later in the development of osteichthyan fishes (*Figure 7*). Based on extensive developmental series of different vertebrate taxa, we present a new interpretation of some lungs previously considered as paired, and therefore, a new definition of paired lungs. Based on our results, true paired lungs are stated when bilateral lung buds arise simultaneously and are both connected directly to the foregut, as observed in the salamander (*Figures 5B, D and 7*).

The sister group to all other living actinopterygian (polypterids) and all living sarcopterygian fishes have a clearly unpaired lung in early developmental stages (*Figure 1B*) that can be developed in later stages either in a unilobed or a secondarily multilobed lung (*Figures 1D–F , and 7*), but not in a true paired lung. *P. senegalus* and the lungfish *L. paradoxa* possess secondary multilobed structure from the larval stage onward since the lung is derived from a unilateral connection to the foregut (*Figure 4G, H, J and K*). The presence of this secondary multilobed morphology likely represents an advantage for the obligatory air-breathing behavior of these taxa, raising the respiratory compliance. In the teleost *Batrachomoeus trispinosus*, the non-respiratory gas bladder is also described as paired (*Rice and Bass, 2009*), although it is most probably a secondary condition.

The most parsimonious scenario inferred from our data mapped on the phylogenetic framework (*Figure 7*) is that the vertebrate lung was unpaired at the evolutionary origin. Since soft tissues are rarely preserved in fossils, living lunged osteichthyans are key taxa for the understanding of how

evolutionary constraints shaped breathing adaptations on land. Our study suggests that the ancestral condition of the lung is a median unpaired organ (*Figure 7*), thereby being inconsistent with the scenario that the lung evolved through a modification of the posteriormost pharyngeal pouch assumed to be present in primitive taxa (*Kastschenko, 1887*; *Goodrich, 1931*). Consequently, the evolutionary origin of the lung was likely independent of the pharyngeal pouch at the morphological level.

From this evolutionary point of view, our results lead to a new definition of the vertebrate lung: either an unpaired or paired respiratory organ developing ventrally from the foregut. Vestigial forms secondarily released from the respiratory function should be also designated as lungs (e.g. the lung of coelacanths). Some criteria previously used for discriminating lungs from gas bladders are no longer supported, including: paired/unpaired organization, position ventral to the alimentary tract (*Marcus, 1937*; *Funk et al., 2020*; *Lambertz et al., 2015*; *Graham, 1997*), as well as its function. The dorsal position of the majority of osteichthyans lungs described here may be related to its dual and secondary functionality of respiration and buoyancy control (*Thomson, 1968*). Actually, the only morphological characteristic that can be used to distinguish lungs and gas bladders is the ventral and dorsal origins from the foregut, respectively (*Funk et al., 2020*; *Cass et al., 2013*). This phenotypic differentiation into true paired lungs in tetrapods may be related to differential gene expressions (*Funk et al., 2020*; *Bi et al., 2021*). Nevertheless, at a level of developmental mechanism, the possibility of co-options of gene regulatory networks of the pharyngeal pouch morphogenesis cannot be excluded, as both the lung bud and pharyngeal pouch develop through the invagination of the foregut endoderm. Our results open the door for future molecular analyses to trace possible regulatory elements for the evolutionary transition from unpaired lungs to true paired lungs in tetrapods.

According to morphological evidence presented here, bifurcation morphogenesis into true paired lungs was not developed yet in osteichthyan fish ancestors. Based on the extant phylogenetic bracket, we infer that the bilaterally paired nature of the lung evolved only in the lineage towards fossil and extant tetrapods, as a synapomorphy of this clade (*Figures 5 and 7*). This morphological modification brought about improvement of the efficiency in oxygen-uptake from the air, as the paired lungs having parallel air flows exchange the air more quickly than the unpaired lung having only single air flow does. This innovation led to the elevation of metabolic rate that was required for the sustained body support against the gravity. Paired lungs may have been present also in early tetrapods and were probably essential to raise lung surface area and volume capacity during the evolution of vertebrate respiratory system and the air-breathing intensification at the water-to-land transition.

## Materials and methods
### Specimens information

All specimens used in this work are permanently housed in collections of public institutions. No specimens were collected alive in the field for this work. *P. senegalus* specimens were originally obtained for the study on the molecular developmental in polypterids (*Tatsumi et al., 2016*). Nine specimens here studied are: six late embryos (free embryonic phase or postembryos) of 8.0 mm TL (PS-001–01) and histological thin-section of another specimen of 8.0 mm TL (PSS-No1), 8.5 mm TL (PS-001–02), 9.1 mm TL (PSS-No2), and 9.3 mm TL (two specimens, PS-001–03); four larva of 12 mm TL (two specimens, PS-001–04), 15.5 mm TL (PSS-No3), and 18.0 mm TL (PSS-No4); and three juveniles of 20 mm TL (PS-001–05), 23 mm TL (PS-001–06), and 45 mm TL (PS-001–07). We indicate the developmental stages (embryo, larvae, juveniles, and adults) following *Bartsch et al., 1997*. Specimens and histological material are housed at the Department of Anatomy of the Jikei University School of Medicine (Tokyo, Japan).

Four specimens of *L. paradoxa* here studied are from the collections of the Universidade do Estado do Rio de Janeiro and were collected legally in 2008, with the permission number 11471–1. The specimens are registered under the acronym UERJ-PN: UERJ-PN 550 is a larva of 46 mm TL; UERJ-PN 262 is a juvenile of 68 mm TL; UERJ-PN 238 is juvenile of 222.1 mm TL; and PC02 is an adult of 400 mm TL. We follow *Kerr, 1900* for the developmental staging of *Lepidosiren*.

Specimens of *N. forsteri* were collected legally from Department of Biological Sciences, Macquarie University, Sydney, Australia, and transported with the permission of CITES (Certificate No. 2009-AU-564836). The developmental series comprises fourteen specimens. Sizes are: an early embryo

of 13.5 mm TL (IMU-RU-SI-0013); 11 larvae of 16 mm TL (IMU-RU-SI-0017), 17 mm TL (IMU-RU-SI-0019 and IMU-RU-SI-0022), 17.5 mm TL (IMU-RU-SI-0037), 19 mm TL (IMU-RU-SI-0038), 20.5 mm TL (IMU-RU-SI-0039), 24 mm TL (IMU-RU-SI-0040), 25.5 mm TL (IMU-RU-SI-0041), 26.5 mm TL (IMU-RU-SI-0042), 30 mm TL (IMU-RU-SI-0043), and 50 mm TL (IMU-RU-SI-0045); a juvenile of 70 mm TL (IMU-RU-SI-0048); and an adult specimen of 200 mm TL (KPM-NI 11384). For the developmental identification (embryos, hatchlings/larvae, juveniles, and adults) we follow *Kemp, 1982*, *Kemp, 2011* and *Ziermann et al., 2018*.

Six *S. salamandra* specimens were obtained on loan at the amphibian collection of the Muséum national d'Histoire naturelle (Paris, France). The three developmental stages (as described at the MNHN collection) are: two early larvae MNHN 1978.636 (1) of 35.5 mm TL and MNHN 1978.636 (2) of 42.8 mm TL; larva MNHN 1985.9039 of 49.6 mm; larva in metamorphosis MNHN 1978.542 of 54.44 mm TL; small juvenile MNHN 1988.7177 of 50 mm TL; juvenile MNHN 1962.1004 of 81.85 mm TL.

Institutional abbreviations: IMU-RU-SI, Iwate Medical University, Ryozi Ura Collection, Japan; PS, *Polypterus senegalus*; PSS, *Polypterus senegalus* sections; KPM-NI, Kanagawa Prefectural Museum Natural History, Odawara, Japan; MNHN, Muséum national d'Histoire naturelle, Paris, France; UERJ-PN, Universidade do Estado do Rio de Janeiro, Peixes Neotropicais.

## X-ray tomography

Due to the extremely small size of the embryos and larvae, and to the weak density difference between soft tissues of the abdominal cavity, propagation phase-contrast microtomography was the unique way to study their anatomy and histology at micrometer scale. Phase-contrast microtomography being only achieved at synchrotron sources, we accessed the anatomy of these rare and tiny samples at the Synchrotron SOLEIL and Synchrotron SPring-8. The high brightness of the synchrotrons was essential for our material and enabled the collection high resolution scans in short timescales.

Specimens of *P. senegalus*, *L. paradoxa,* and *S. salamandra* were imaged at the PSICHÉ beamline of the SOLEIL Synchrotron (Saint-Aubin, France) while *N. forsteri* specimens were scanned at SPring-8 Synchrotron. The specimens were scanned isolated in a plastic tube filled with Phosphate-buffered saline (PBS) for *P. senegalus* and *N. forsteri*, ethanol for *L. paradoxa,* and formaldehyde for *S. salamandra*. They were immobilized in vertical position using gauze pads, and/or sank inside the tip of a plastic pipette in the case of tiny individuals, in order to benefit as much possible from the available field of view and thus achieve the highest possible resolution.

At SOLEIL Synchrotron, imaging was performed using a monochromatic beam with an energy of 25 keV. A series of acquisitions with vertical movement of the sample were recorded to extend vertically the field of view and image the entire (or most of the) individual. Two distinct setups were used to accommodate the different sizes of the individuals (size variations occurring both between developmental stages and taxa). (1) Small individuals were scanned using a field of view of $2.6 \times 2.6$ mm$^2$ ($5\times$ magnification) resulting in a projected pixel size of 1.3 µm, and a propagation distance of 148 mm. (2) Larger individuals were scanned using a field of view of ~$12.6 \times 3.3$ mm$^2$ ($1\times$ magnification) resulting in a projected pixel size of 6.17 µm, and a propagation distance of 500 mm. For individuals slightly wider than these field of views, the latter were extended horizontally by positioning the rotation axis off-centre and acquiring data over a 360° rotation of the sample. The volumes were reconstructed from the (vertically) combined radiographs using PyHST2 software (*Mirone et al., 2014*), with a Paganin phase retrieval algorithm (*Paganin et al., 2002*). The huge resulting volumes (from 70 Gb to 1.2 Tb) were reduced (crop, rescale 8-bit, binning) to facilitate 3D data processing.

Specimens of *N. forsteri* (from 13.5 to 70 mm TL) were imaged at the SPring-8 Synchrotron, beamline 20B2. For specimens from 13.5 mm TL to 30 mm TL, a beam energy of 15 keV was used with a double bounce Si (111) monochromator. Data were obtained at three different resolutions, and correspondingly used three combinations of two lenses and fluorescent material, as follows, 2.75 µm/voxel; 1st-stage lens: 'beam monitor 2' f35 mm; 2nd-stage lens: Nikon 85 mm lens; GADOX thickness: 15 µm 4.47 µm/voxel; 1st-stage lens: 'beam monitor 2' f35 mm; 2nd-stage lens: Nikon 50 mm lens; GADOX thickness: 15 µm 12.56 µm/voxel; 1st-stage lens: 'beam monitor 5' f200 mm; 2nd-stage lens: Nikon 105 mm lens; GADOX thickness: 25 µm.

Datasets were acquired at propagation distances of 2.75 µm/voxel, 4.47 µm/voxel: 600 mm; 12.56 µm/voxel: 3 m and three different exposure times of 70 ms, 150 ms, and 200 ms per projection.

Field of view were: pixel size × 2048 (2.75 × 2048 = 5632 μm; 4.47 × 2048 = 9154.56 μm; 12.56 × 2048 = 25722.88 μm) A total of 1800 projections were recorded per scan as the sample was rotated through 180°. A high-resolution computerized axial tomography scanning was performed for the adult specimen of *Neoceratodus* (KPM-NI 11384) of 200 mm TL at the National Museum of Nature and Science (Tokyo, Japan) using the following scanning parameters: effective energy 189 kV, current 200 mA, voxel size 9.765 μm and 1000 views (slice width 0.1 mm).

## Segmentation and three-dimensional rendering

Segmentation and 3D rendering were performed using the software MIMICS Innovation Suite 20.0 (Materialise) at the Laboratório de Ictiologia Tempo e Espaço of the Universidade do Estado do Rio de Janeiro.

## Acknowledgements

We are especially indebted to Anthony Graham, John A Long, as well as an anonymous referee for their valuable comments. We thank Dr Annemarie Ohler (curator of the Reptiles and Amphibians Collection of the Muséum national d'Histoire naturelle) for the loan of *Salamandra* specimens and J Joss (Macquarie University) for providing *Neoceratodus* embryos. We are grateful to Dr H Seno (curator of the Ichthyology Collection of the Kanagawa Prefectural Museum of Natural History) for the loan of *Neoceratodus* specimen KPM-NI 11384. Funding: Coordenação de Aperfeiçoamento de Pessoal de Nível Superior—Brasil (CAPES)—grant Finance Code 001 (Programa Nacional de Pós Doutorado-PNPD) (CC) Programa de Apoio à Docência (PAPD) grant (E-26/007/10661/2019)— Universidade do Estado do Rio de Janeiro (CC) Interdisciplinary Collaborative Research Program of the Atmosphere and Ocean Research Institute, the University of Tokyo JP (CC, TS) Prociência fellowship CNPq grant 310101/2017–4 (PMB) FAPERJ grants E-26/200.605/2022 (CC); E-26/210.369/2022 (CC); E-26 /202.890/2018 (PMB)

## Additional information

### Funding

| Funder | Grant reference number | Author |
|---|---|---|
| Coordenação de Aperfeiçoamento de Pessoal de Nível Superior | Finance Code 001 | Camila Cupello |
| Programa de Apoio à Docência | E-26/007/10661/2019 | Camila Cupello |
| Fundação Carlos Chagas Filho de Amparo à Pesquisa do Estado do Rio de Janeiro | E-26/200.605/2022 | Camila Cupello |
| Fundação Carlos Chagas Filho de Amparo à Pesquisa do Estado do Rio de Janeiro | E-26/210.369/2022 | Camila Cupello |
| Fundação Carlos Chagas Filho de Amparo à Pesquisa do Estado do Rio de Janeiro | E-26/ 202.890/2018 | Paulo M Brito |
| Interdisciplinary Collaborative Research Program of the Atmosphere and Ocean Research Institute, the University of Tokyo | | Camila Cupello Toshiro Saruwatari |
| Prociência Fellowship CNPq | 310101/2017-4 | Paulo M Brito |

| Funder | Grant reference number | Author |
|--------|------------------------|--------|

The funders had no role in study design, data collection and interpretation, or the decision to submit the work for publication.

## Author contributions

Camila Cupello, Conceptualization, Data curation, Formal analysis, Funding acquisition, Investigation, Methodology, Project administration, Resources, Software, Supervision, Validation, Visualization, Writing – original draft, Writing – review and editing; Tatsuya Hirasawa, Data curation, Formal analysis, Investigation, Methodology, Software, Validation, Visualization, Writing – original draft, Writing – review and editing; Norifumi Tatsumi, Data curation, Formal analysis, Investigation, Methodology, Resources, Software, Validation, Visualization, Writing – original draft, Writing – review and editing; Yoshitaka Yabumoto, Conceptualization, Data curation, Funding acquisition, Investigation, Methodology, Software, Validation, Visualization, Writing – original draft, Writing – review and editing; Pierre Gueriau, Data curation, Investigation, Methodology, Software, Validation, Visualization, Writing – original draft, Writing – review and editing; Sumio Isogai, Ryoko Matsumoto, Data curation, Methodology, Validation, Writing – review and editing; Toshiro Saruwatari, Data curation, Funding acquisition, Validation, Writing – review and editing; Andrew King, Masato Hoshino, Kentaro Uesugi, Methodology, Software, Validation, Writing – review and editing; Masataka Okabe, Data curation, Investigation, Methodology, Validation, Writing – original draft, Writing – review and editing; Paulo M Brito, Conceptualization, Data curation, Funding acquisition, Methodology, Validation, Writing – original draft, Writing – review and editing

## Author ORCIDs

Camila Cupello ⓘ http://orcid.org/0000-0001-6030-8670
Tatsuya Hirasawa ⓘ http://orcid.org/0000-0001-6868-3379
Norifumi Tatsumi ⓘ http://orcid.org/0000-0001-6640-150X
Yoshitaka Yabumoto ⓘ http://orcid.org/0000-0002-4736-8246
Pierre Gueriau ⓘ http://orcid.org/0000-0002-7529-3456
Ryoko Matsumoto ⓘ http://orcid.org/0000-0002-0334-7259
Toshiro Saruwatari ⓘ http://orcid.org/0000-0002-5761-8528
Andrew King ⓘ http://orcid.org/0000-0001-8542-1354
Masato Hoshino ⓘ http://orcid.org/0000-0002-4167-2706
Kentaro Uesugi ⓘ http://orcid.org/0000-0003-2579-513X
Masataka Okabe ⓘ http://orcid.org/0000-0002-8618-6859
Paulo M Brito ⓘ http://orcid.org/0000-0002-4853-8630

## Decision letter and Author response

Decision letter https://doi.org/10.7554/eLife.77156.sa1
Author response https://doi.org/10.7554/eLife.77156.sa2

# Additional files

## Supplementary files

• Transparent reporting form

## Data availability

All data used in this work are publicly available via Dryad https://doi.org/10.5061/dryad.vdncjsxx1.

The following dataset was generated:

| Author(s) | Year | Dataset title | Dataset URL | Database and Identifier |
|-----------|------|---------------|-------------|-------------------------|
| Cupello C | 2022 | Data from: Lung evolution in vertebrates and the water-to-land transition | http://dx.doi.org/10.5061/dryad.vdncjsxx1 | Dryad Digital Repository, 10.5061/dryad.vdncjsxx1 |

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
