## [Editor Report]

This study focused on five osteichthyan vertebrate species and investigated their lung morphology. The comparison of the observations suggests an origin of the lung as an unpaired organ, with the present-day paired forms in amniotes being a result of secondary modification. The sound morphological comparison presented provides valuable insight into the evolution of the lung. The work will be of interest to colleagues studying vertebrate evolution.

---

## [Decision Letter]

**Decision letter after peer review:**

Thank you for submitting your article "Lung evolution in vertebrates and the water-to-land transition" for consideration by *eLife*. Your article has been reviewed by 3 peer reviewers, one of whom is a member of our Board of Reviewing Editors, and the evaluation has been overseen by Marianne Bronner as the Senior Editor. The following individuals involved in the review of your submission have agreed to reveal their identity: Anthony Graham (Reviewer #2); John A Long (Reviewer #3).

Essential revisions:

1) The authors are requested to mention what was reported about the placoderm lung and discuss the discrepancy raised by Reviewer #3.

2) The manuscript should be checked for basic grammar, as pointed out by Reviewer #1.

*Reviewer #1 (Recommendations for the authors):*

I have several suggestions for consolidating the value of the manuscript listed below.

1. The authors are suggested to be careful about the use of the word 'basal' – see this literature: https://resjournals.onlinelibrary.wiley.com/doi/10.1111/j.0307-6970.2004.00262.x

2. Line 93 – Did the authors really analyze the 'primitive sequence' of morphogenesis? They analyzed the morphology of present-day animals in different lineages and should not have accessed any 'sequence'.

3. Line 234 'which had evolved secondarily' – This part sounds too definitive and is recommended to indicate that it is just speculation. This suggestion applies to a similar part in Line 327/328.

4. The authors are recommended to refer to the individual figures (not only to Figure 7 for summary) in the Discussion.

5. Line 313 – develop -> develops.

6. Line 320 – clear unpaired -> clearly unpaired(?).

7. Line 331 – tissue -> tissues.

8. The sentence in Line 338-340 is difficult to follow and is recommended to be rewritten.

9. Line 333 – revealed -> suggested.

10. Line 341- The word 'useful' sounds too casual, so this part needs to be rewritten.

11. Line 350 – the phrase 'developmental genetic level' sounds awkward, so this part needs to be rewritten.

12. Line 357-359 – This statement is limited to modern species and ignores extinct evolutionary lineage. I suggest rewriting this sentence.

13. Line 366 – 'water-to-land' with hyphens.

*Reviewer #2 (Recommendations for the authors):*

The Latimeria chalumnae data have been previously published – this should be more clearly flagged in the paper.

*Reviewer #3 (Recommendations for the authors):*

This paper by a leading team of established experts in basal osteichthyan respiratory anatomy and evolution provides significant new data on the anatomy and developmental biology and evolution of lungs in key basal, key osteichthyan fishes and a primitive tetrapod. The paper presented a sound historical background to the problem, but a few points that should be considered are not included. The 3D CT data combined with thin-section images substantiate their findings well. The work holds important implications for understanding how paired lungs first evolved in fishes and tetrapods, which has been a major evolutionary conundrum up to now. The methods and data presented are sound, and the illustrations are clear and relevant to the development of the intellectual arguments presented in the discussion.

---

## [Author Response]

Reviewer #1 (Recommendations for the authors):I have several suggestions for consolidating the value of the manuscript listed below.1. The authors are suggested to be careful about the use of the word 'basal' – see this literature: https://resjournals.onlinelibrary.wiley.com/doi/10.1111/j.0307-6970.2004.00262.x

We have removed the unnecessary uses of the word ‘basal’ (lines 53 and 188 in the first version of the manuscript, now lines 62 and 107 respectively).

2. Line 93 – Did the authors really analyze the 'primitive sequence' of morphogenesis? They analyzed the morphology of present-day animals in different lineages and should not have accessed any 'sequence'.

We have accordingly deleted the mention to “primitive sequence” (now line 112).

3. Line 234 'which had evolved secondarily' – This part sounds too definitive and is recommended to indicate that it is just speculation. This suggestion applies to a similar part in Line 327/328.

We agree with the reviewer’s comment and have tempered our statements line 234 and 327/328 (now lines 282 and 409, respectively).

4. The authors are recommended to refer to the individual figures (not only to Figure 7 for summary) in the Discussion.

Our Discussion section now includes several references to individual figures instead of only to Figure 7.

5. Line 313 – develop -> develops.

Change has been done.

6. Line 320 – clear unpaired -> clearly unpaired(?).

It should have indeed stated “clearly unpaired”, which we corrected accordingly.

7. Line 331 – tissue -> tissues.

Change has been done.

8. The sentence in Line 338-340 is difficult to follow and is recommended to be rewritten.

We have written this sentence as follows (now lines 420-423):

"From this evolutionary point of view, our results lead to a new definition of the vertebrate lung: either an unpaired or paired respiratory organ developing ventrally from the foregut. Vestigial forms secondarily released from the respiratory function should be also designated as lungs (e.g., the lung of coelacanths)."

9. Line 333 – revealed -> suggested.

Changes have been done.

10. Line 341- The word 'useful' sounds too casual, so this part needs to be rewritten.

We have replaced “useful” with the more appropriate term “supported” (now line 427).

11. Line 350 – the phrase 'developmental genetic level' sounds awkward, so this part needs to be rewritten.

Although we don't consider the phrase 'developmental genetic level' to be strange (the similar usage has been appeared elsewhere, for examples, Wagner, 2007: Nature Reviews Genetics; Kitazawa et al., 2015: Nature Communications), we have replaced “developmental genetic level” with the “a level of developmental mechanism” (now lines 436, 437).

12. Line 357-359 – This statement is limited to modern species and ignores extinct evolutionary lineage. I suggest rewriting this sentence.

We have written this sentence as follows (now lines 444-447):

“Based on the extant phylogenetic bracket, we infer that the bilaterally paired nature of the lung evolved only in the lineage towards fossil and extant tetrapods, as a synapomorphy of this clade (Figures 5 and 7).”

13. Line 366 – 'water-to-land' with hyphens.

Corrected (now line 454).

Reviewer #2 (Recommendations for the authors):The Latimeria chalumnae data have been previously published – this should be more clearly flagged in the paper.

We agree and have added the following sentence to make this point clearer in the text (lines 206-209 of our revised version):

“The anatomy of the lung of fossil and extant coelacanths, as well as its ontogenic development in the extant *Latimeria chalumnae*, have been extensively documented (Cupello et al., 2015; Cupello et al., 2017a; Cupello et al., 2017b; Cupello, Clément and Brito, 2019).”